# Extracellular Vesicles as Delivery Vehicles for Therapeutic Nucleic Acids in Cancer Gene Therapy: Progress and Challenges

**DOI:** 10.3390/pharmaceutics14102236

**Published:** 2022-10-19

**Authors:** Rong Du, Chen Wang, Ling Zhu, Yanlian Yang

**Affiliations:** 1CAS Key Laboratory of Standardization and Measurement for Nanotechnology, National Center for Nanoscience and Technology, Beijing 100190, China; 2University of Chinese Academy of Sciences, 19 A Yuquan Rd., Shijingshan District, Beijing 100049, China; 3CAS Key Laboratory of Biological Effects of Nanomaterials and Nanosafety, National Center for Nanoscience and Technology, Beijing 100190, China

**Keywords:** extracellular vesicles, engineering, therapeutic nucleic acids, cancer therapy

## Abstract

Extracellular vesicles (EVs) are nanoscale vesicles secreted by most types of cells as natural vehicles to transfer molecular information between cells. Due to their low toxicity and high biocompatibility, EVs have attracted increasing attention as drug delivery systems. Many studies have demonstrated that EV-loaded nucleic acids, including RNA-based nucleic acid drugs and CRISPR/Cas gene-editing systems, can alter gene expressions and functions of recipient cells for cancer gene therapy. Here in this review, we discuss the advantages and challenges of EV-based nucleic acid delivery systems in cancer therapy. We summarize the techniques and methods to increase EV yield, enhance nucleic acid loading efficiency, extend circulation time, and improve targeted delivery, as well as their applications in gene therapy and combination with other cancer therapies. Finally, we discuss the current status, challenges, and prospects of EVs as a therapeutic tool for the clinical application of nucleic acid drugs.

## 1. Introduction

Cancer is one of the most significant burdens a person can bear. Cancer is responsible for one out of every six deaths worldwide, according to the World Health Organization [1]. Most traditional antitumor small molecule chemotherapeutics and antibody drugs work by binding to target proteins, but the druggability of the target proteins limits their development. Only 3000 of the roughly 20,000 proteins encoded by the human genome are druggable, and only 700 have corresponding drugs in development [2,3]. Nucleic acid drugs can modulate extracellular and cell membrane proteins, whereas antibody drugs only act on cell membranes and extracellular proteins.

Cancer is caused by changes in genetic material, such as genetic mutations and chromosomal aberrations, which eventually lead to continued proliferation and metastasis. Nucleic acid drug therapy can begin at the source of the disease by exogenously introducing the therapeutic genes into diseased cells, correcting the disease caused by gene defect and abnormality, and achieving a therapeutic effect on the tumor [4]. Because of its flexibility and targeting, cancer gene therapy is emerging as a promising therapy for treating tumors [5,6]. The sequence of the nucleic acid drug can be easily designed using the complementary base pairing principle and knowing the base sequence of the target genes. Gene drugs can achieve breakthroughs in difficult-to-make protein targets and have a high potential for developing drugs for “untargetable” and “undruggable” diseases [7].

Despite the great attention paid to nucleic acid drugs, the clinical application of gene therapy is still limited by the current inefficient delivery of these molecules to target cells. For nucleic acid drugs to enter the body, several problems must be overcome: First, they need to overcome a billion-year-old defense mechanism that cells have evolved to prevent external RNA from invading the cell’s interior [8]. Second, naked nucleic acids are easily degraded by RNase enzymes in plasma and tissues and are rapidly cleared by the liver and kidney and recognized by the immune system [9,10]. Therefore, efficient delivery of RNA into cells remains a major issue for the widespread development of RNA therapeutics, and suitable carriers need to be found.

Extracellular vesicles (EVs), which are cell-derived, phospholipid-based bilayer membrane particles, are considered potential bioderived nanocarriers. Compared with synthetic lipid nanoparticles, EVs have natural biological advantages: low toxicity, low immunogenicity, exudation in tissues, ability to cross biological barriers, targeting of specific cell types, easy fusion with cell membranes, and ability to achieve endo/lysosomal escape. Moreover, nucleic acid drugs loaded in EVs are naturally protected from circulation degradation, which is a major advantage of EVs as drug delivery systems (DDS) [11,12,13].

In this context, this article aims to review several methods for improving EV drug delivery systems, as well as examples and recent advances in EV delivering nucleic acid drugs for cancer therapy.

## 2. EVs as Nucleic Acid Drug Delivery Vehicles

The International Society for Extracellular Vesicles (ISEV) is a professional social group composed of researchers and scientists in the field of EVs. It is committed to promoting global EV research and is one of the most authoritative societies in the field of EVs. As defined by the ISEV, EVs are the general term for particles that are naturally released from cells, which are separated by lipid bilayers and cannot replicate, i.e., do not contain a functional nucleus [14]. They can be endosome-derived (termed exosomes, diameter 30~150 nm) or are generated by membrane outward budding (termed ectosomes, diameter 50~1000 nm) [15]. From either biogenesis, EVs are natural vehicles carrying and transferring biological information for cellular communication and have attracted increasing attention as drug delivery systems. Typically, EVs’ surface proteins reflect intracellular markers, such as LAMP1, LAMP2b, and ALIX-1 proteins, and express the tetraspanins CD9, CD63, and CD81 [16]. In living systems, EVs are used to transmit biological signals, deliver proteins and nucleic acids, and induce various biological effects, such as mediating tumor metastasis [17]. Based on this, EVs can be applied in targeted therapy, cell-free therapy, and drug delivery systems [18,19].

Currently, the most widely used nucleic acid drug delivery carriers are viruses and lipid nanoparticles (LNPs). Nearly 70% of clinical studies have used viral vectors [20,21], but a major limitation is an immune rejection and uncontrollable side effects [22,23,24]. Although LNPs are the most widely used nano-delivery systems for RNA-based therapy [25], due to the intrinsic mechanism of LNPs, innate immune recognition sensors are often triggered, resulting in acute hypersensitivity reactions, and hindering their clinical application in hypersensitive patients. In contrast, EVs are more stable, less immunogenic, less toxic, and well-tolerated in humans than LNPs [26]. EVs consist of a natural mixture of biomolecules that do not cause the adverse effects associated with liposomal particle infusion. Several clinical trials using EVs for immunotherapy demonstrate EVs’ safety in humans [27].

Many types of cells are suitable for producing EVs based on their natural properties. Stem cells are favored for their high safety and high EV secretion and have been used in clinical studies [28]. Group O-red blood cells can be used as universal donors for large-scale EV production because they are readily available in blood banks and do not contain DNA [29]. Immune cells such as macrophages and dendritic cells are also commonly used because their EVs express specific signaling molecules that evade clearance by the immune system [30,31]. In the field of tumor therapy, EVs secreted by homologous tumor cells are widely used because of the homing effect on homologous tumors and can achieve active targeting [32,33,34]. However, as a drug delivery system, the following aspects still need to be considered: (1) EV secretion varies widely among different cell types and subpopulations [35] and may be further influenced by cell state and growth conditions [30]. Since various types of cells can generate EVs in response to endogenous or exogenous stimuli, how to improve the production of EVs is a key step for the widespread application and industrialization of EVs as DDS [36]. (2) Improving the encapsulation rate of nucleic acid drugs is also a consideration for realizing industrialization. (3) Although EVs themselves can circumvent the clearance of the mononuclear phagocytic system to a certain extent (clearing circulating particles larger than 100 nm), engineering modifications are required to maximize their circulation time and emphasize their advantages in intercellular communication [37]. (4) The different characteristics of EV producers and target cells may lead to significant differences in the efficiency of cell-to-cell communication. The efficiency of cellular uptake may be affected by surface-specific proteins, lipopolysaccharide decoration, and the overall potential (usually negative charge) of EVs. Therefore, targeting modifications for EVs have been extensively studied [38,39]. Given the foregoing, it is critical to design EVs to improve the efficiency and quality of nucleic acid drug delivery vehicles.

## 3. Improvements in EV Drug Delivery Systems

EVs are nano-scale vesicles with surfaces composed of a heterogeneous mixture of lipids and proteins, and naturally have the advantages of stealth, biocompatibility, and intrinsic homing ability. Although natural EVs already have certain targeting, long-term circulation, and cell entry capabilities, researchers are not limited to using natural EVs but intend to engineer them for better effects [5,40,41]. Here, we discuss improvement strategies according to the following four purposes (Figure 1): (A) to increase EV production; (B) to improve nucleic acid drug loading efficiency; (C) to extend circulation time; (D) to improve targeting capability and introduce corresponding modification methods (Table 1). At the end of this section, we compare the advantages and disadvantages of various strategies for modification purposes (Figure 2).

### 3.1. To Increase EV Production

The low yield of EV secretion is a significant barrier to large-scale production, limiting its clinical potential as a drug delivery platform. Improving EV yield can begin with changing the cell culture mode, inducing EV secretion via Ca^2+^-dependent regulation, applying different external stimuli to cells under culture conditions, and finally improving EV separation methods.

#### 3.1.1. Changing Cell Culture Methods

Although high-density culture can be achieved in suspension cells, for adherent cells, high-density culture is difficult due to the occurrence of contact inhibition and results in low numbers of EVs for subsequent studies. The Integra CELLine Culture System is a two-compartment culture flask with a semipermeable membrane surrounding a concentrated cell-containing compartment, allowing a constant source of nutrients from a larger external culture compartment. Compared to commonly used flask systems (0.78 μg/mL medium), CELLine is less costly and time-consuming and increases EV yield by nearly 12-fold (10.06 μg/mL medium) [44]. Using this culture method, the researchers obtained 37.2 μg/mL EVs from the culture supernatant of PANC-1 cells [42]. Vertical-Wheel^TM^ Bioreactor (VWBR), developed by PBS biotech^®^, utilizes vertically rotating flow channels to promote radial and axial fluid flow to create a more homogeneous hydrodynamic environment. It has been applied to mesenchymal stromal cells (MSC), resulting in increased EVs production by 5.7 ± 0.9 times [45].

The researchers experimented with three-dimensional culture to increase the yield of EVs and maximize the culture areas [47]. Watson et al. [46] designed a hollow fiber bioreactor for culturing cells within the fibers of the device, demonstrating that the hollow fiber system can increase the EV yield by 5–10-fold and reduce contaminants, making it ideal for large-scale production. Similarly, Cao et al. [43] found by the same method that the yield of EVs extracted from 3D culture was 19.4 times higher than that of 2D culture and did not affect the phenotype of MSC. Microcarrier suspension culture is by far the most suitable platform for 3D stem cell culture, and the researchers noted that when this system was applied, the production of EVs from Wharton’s jelly (i.e., umbilical cord connective tissue)-derived mesenchymal stem cells could be further increased by about 20 times [58].

#### 3.1.2. Changing EV Separation Methods

Based on particle density and size, ultracentrifugation is the gold standard method for EV isolation currently in use. However, this method is time-consuming and expensive, and the particles may contain non-EV contaminants, particularly lipoprotein complexes with similar densities, making it difficult to adapt to large-scale homogenization production [47]. Other methods, such as size exclusion chromatography, ultrafiltration, and immunoaffinity methods, although higher in purity, are limited to EV extraction from small amounts of samples and are also not suitable for large-scale production [85].

Tangential flow filtration (TFF) technology is a scalable concentration and buffer exchange strategy for the large-scale production of biologics. The flow direction in the TFF is perpendicular to the filtration direction. Because of this, this method can effectively avoid the formation of filter cake on the membrane surface and improve the membrane’s utilization rate and the equipment’s stability [60]. Compared with conventional ultracentrifugation (UC) in 2D cell culture, TFF increased the EV yield by 27-fold. The cumulative effect of TFF and 3D culture (3D-TFF-exosome) resulted in a 140-fold increase in EV production compared to 2D-UC-exosome [58]. Kim et al. [59] compared ultracentrifugation and TFF-based separation methods. The results confirmed that the TFF-based isolation method increased the EV yield by two orders of magnitude compared to the ultracentrifugation method.

#### 3.1.3. Inducing EV Secretion by Ca^2+^-Dependent Regulation

EV secretion is dependent on cytoskeleton reorganization and is regulated by calcium-dependent mechanisms. According to previous reports, K562 cells were stimulated with monensin to significantly increase calcium ion concentration. Since monensin induces Ca^2+^ entry by the reversed activity of the Na^+^/Ca^2+^ exchanger, resulting in an initial increase in cytosolic Ca^2+^, it was demonstrated that Rab11 in the K562 leukemia cell line could regulate calcium-induced EV secretion [48]. Subsequent reports suggested that the mechanism of inducing EV release was calcium ion and small molecule RhoGTPase regulation, affecting downstream proteins such as Rab5b and related Rab family proteins [51]. In malignant melanoma cells, Rab27a is involved in EV formation. Reducing the level of Rab27a in cells results in a 50% reduction in EV secretion [49]. Munc13-4 is a calcium-dependent SNAP receptor and Rab-binding protein that is required for calcium-dependent membrane fusion. Munc13-4 uses a Rab11-dependent transport pathway to generate a multivesicular body (MVB) capable of releasing EVs, and deletion of Munc13-4 reduces the size of CD63^+^ MVBs, suggesting a role for Munc13-4 in MVB maturation. Signal transduction mechanisms may be involved in the activation of EV-derived cells to release these small vesicles in place [50]. As a result, increasing calcium ions to form and release EVs can stimulate the secretion of more EVs.

#### 3.1.4. Inducing EV Secretion by Stressed Culture Conditions

Under hypoxic conditions, cancer cells may release more EVs into their microenvironment to promote their survival and invasion [54]. In hypoxia, HIF-1 mediates the generation of EVs in a time-dependent manner. Experiments have shown that the increase in HIF-1 induced by dimethoxy glycine also promotes EV secretion, while the inhibition of HIF-1 by drugs and genes can inhibit the increase in EV secretion. Under hypoxic stimulation, renal tubular epithelial cells can promote the production and secretion of EVs and participate in protecting EVs in renal tubular cells [53]. Since tumor cell-derived EVs may be used as a delivery system for the paracrine proliferation of tumor malignancies, the low pH characteristic of the tumor microenvironment has also been shown to enhance EV increase [55]. Likewise, cardiac stem cells under electrical stimulation may release more EVs with higher content of protective molecules, promoting cardiomyocyte survival and angiogenesis due to a protective mechanism for cells [56]. Z. Yang et al. [52] designed a cellular nanoporation (CNP) device to stimulate cells with localized and transient electrical pulses to facilitate the release of EVs. The reasons for the increased EV release are as follows: first, the increase in intracellular calcium ions caused by CNP treatment causes EV secretion; second, after CNP treatment, the temperature near the nanochannels increases instantly (<1 s), increasing EV secretion as part of the cellular recovery process caused by the production of heat shock proteins (HSPs).

Liposome stimulation may be a useful strategy for increasing EV yield, and stimulation of cationic naked liposomes without PEG modification has been reported to promote cellular EV secretion; however, their stimulating effect is diminished after liposome modification with PEG. The specific mechanism could be that after being stimulated by liposomes, P53 in tumor cells causes EV secretion by activating purinergic receptors, changing intracellular calcium levels, or causing cell membrane depolarization. The stimulatory/inhibitory effects of liposomes are determined by their dose, surface charge, membrane fluidity, and PEGylation modifications, as well as the type and viability of the cancer cells being treated [86].

In hepatocellular carcinoma cells, stimulation with antitumor drugs significantly increases the production of HSP60, HSP70, and HSP90, especially drug-resistant antitumor drugs, such as irinotecan or carboplatin. The addition of HepG2 drug-resistant drugs caused cells to release more EVs, indicating that heat shock proteins are positively correlated with the EV release [54]. Platinum nanoparticles (PtNPs) provide a promising reagent for enhancing EV yield from A549 cells. PtNPs promote EV release due to oxidative stress and induction of the ceramide pathway [57]. These findings suggest new approaches for promoting EV release.

### 3.2. To Improve Nucleic Acid Drug Loading Efficiency

Common encapsulation methods include pre-transfecting cells with a plasmid vector expressing the target nucleic acid so that EVs continuously secreted by cells encapsulate the nucleic acids [11,87,88]. For the extracted EVs, a common method such as electroporation is to use transient electrical pulses to form pores in membranes to help nucleic acid molecules quickly enter the EV’s cavity, without introducing extracellular molecules, which is safe and non-toxic [62], and the loading rate is usually about 20% [89]. However, it has been found that insoluble siRNA aggregates are massively formed after electroporation, possibly because the discharge in the electroporation dish containing metal electrodes leads to the release of metal cations (such as Al-cations and Fe-cations) from the electrodes [90]. The complex formation of electrode metal ions with hydroxide ions in the electroporation buffer results in siRNA precipitation, so the nucleic acid loading rate may be overestimated. To address the problems caused by electroporation, the researchers found that EDTA acts as a complexing agent and can form soluble complexes with aluminum ions, and studies have shown that adding 1 mM EDTA to the electroporation buffer can significantly reduce siRNA precipitation by 98–99% [91]. Although the number of aggregates caused by sonication is about 12 times less than that induced by electroporation [92], since ultrasound is often used to disrupt lipids or cell membranes, sonication may cause irreversible damage to the integrity of lipid-based EV membranes, resulting in the deformation of EVs, which is not conducive to quality control in mass production.

#### 3.2.1. Physical Strategies

Z. Yang et al. [52] developed a CNP biochip to stimulate cells to produce and release EVs containing target nucleotide sequences. This system allows for the cultivation of monolayers of derived cells, such as mouse embryonic fibroblasts (MEFs) and dendritic cells (DCs), on the surface of a chip containing nanochannel arrays. Nanochannels (approximately 500 nm in diameter) allow transient electrical pulses to pass through, shuttling the PTEN-expressing plasmid from the buffer into the attached cells. Compared with bulk electroporation and other EV production strategies, cellular nanoporation can generate up to 50 times more EV.

Jeyaram et al. [61] reported that the generation of a pH gradient across EV membranes by protonation of EVs can be used to enhance vesicular loading of nucleic acid cargoes, especially short-chain nucleic acids such as microRNAs (miRNAs). Since nucleic acids are negatively charged, acidic pH gradient conditions favor nucleic acid loading. EVs were isolated from HEK293T cells, dehydrated, and then incubated in citrate buffers at pH 2.5 for 1 h. Then, EVs were dialyzed against HEPES buffered saline at pH 7.0 for 24 h to remove free acidic ions, and the modified EVs were incubated with miR-93 in PBS at 22 °C to achieve the highest encapsulation loading of 8 pmol/μg EVs and the loading process did not impair cellular uptake by EVs or promote any significant toxic responses in mice. The authors also show that this loading method allows repeated use of siRNA solutions, as siRNA is not damaged like electroporation and sonication.

For other means, Zhao et al. [93] encapsulated ethylamine-modified bovine serum albumin-coated s100A4 siRNA (CBSA/siS100A4) into exosomal membranes by incubation and extrusion. This approach greatly improved the siRNA encapsulation efficiency (EE), which was measured to be 86.70 ± 1.22%.

#### 3.2.2. Chemical Strategies

Munagala R. et al. [94] used functionalized EVs modified with folic acid, followed by interaction with polyethyleneimine, and incubated with siRNA to form the complex called EPM. Compared to conventional methods electroporation (<5%) and chemically transfected Exo-Fect™ (~35%), EPM resulted in significantly (*p* < 0.001) higher siRNA (>90%) entrapment. Hydrophobic modification of RNA molecules is also one way. Co-incubation can efficiently load hydrophobically modified small interfering RNA (hsiRNA) without disrupting the size distribution and integrity of EVs [62]. The cholesterol moiety is attached to the 3′ end of the guest strand, enabling rapid membrane binding, and the single-chain phosphorothioate promotes cellular internalization by a mechanism like that of traditional antisense oligonucleotides. After incubation, there are approximately 1000 to 3000 hsiRNAs per EV, approaching saturation at 3000 [64]. The group subsequently studied the use of triethyl glycol (TEG) linkers to couple hsiRNA to cholesterol, and the results showed that cholesterol-TEG-hsiRNAs were more efficiently loaded onto extracellular vesicles [63]. This simple co-incubation of hsiRNAs with EVs is an efficient and gentle method for producing and controlling EV loading with chemically synthesized oligonucleotide cargo. AJ O’Loughlin, et al. [62] co-incubated cholesterol-conjugated small interfering RNAs (cc-siRNAs) with EVs for self-association and explored various co-incubated conditions, including temperature, incubation time, volume, and ratio. When EVs were incubated with cc-siRNA at a ratio of 30 cc-siRNA molecules per EV for 1 h at 37 °C, the loading rate was the highest at 74% when the reaction volume reached 100 μL.

#### 3.2.3. Biological Strategies

To encapsulate more drugs, on the one hand, the particle size of EVs can be enlarged to accommodate more drugs, such as fusion modification with liposomes, and still maintain at the nanoscale to ensure the properties of nanomedicines [65]. Crucially, this way of increasing the load is comparable to methods such as sonication and electroporation without damaging the labile nucleic acid cargoes.

On the other hand, the encapsulation rate of nucleic acids can be improved by biological means. Using RNA fusion proteins, Z. Li et al. [66] fused the exosomal membrane protein CD9 with the RNA-binding protein human antigen R (HuR), which has a relatively high affinity for miR-155. The fused CD9-HuR successfully enriched miR-155 into EVs when miR-155 was overexpressed and achieved an encapsulation rate of 98.2 ± 13.6 copies per modified EV.

### 3.3. To Extend Circulation Time

Macrophages associated with the mononuclear phagocytic system organ are mainly responsible for the rapid clearance and retention of EVs, which severely limits the accumulation of EV particles within target tissues and the release of therapeutic cargo in recipient target cells to exert their intended biological effects [95]. Although EV-based therapy has been shown to slow disease progression, the insufficient residence time of exogenous EVs in circulation may impede clinical translation. Surface modification of EVs to avoid detection by the immune system is a viable strategy to inhibit the removal of EVs and improve the delivery efficiency of their targeted content. The circulating half-life of therapeutic EVs was extended by coating them with various antiphagocytic molecules, which increased their bioavailability to target tissues, transferred therapeutic molecular cargoes, and improved delivery efficacy [40].

#### 3.3.1. Immune Checkpoint Strategies

CD47, originally known as integrin-associated protein (IAP), is a highly glycosylated cell surface protein of the immunoglobulin superfamily, which is expressed in most cells in the body, and interacts with the immunosuppressive receptor signal regulatory protein α (SIRPα) mainly expressed in neurons, dendrites cells and macrophages [96,97]. In the work of Kamerkar et al. [67], Du J. et al. [68], and Cheng L. et al. [69], we can find that by introducing a CD47-overexpressing eukaryotic expression vector into EV-derived cells, CD47 surface functionalization enables EVs to effectively escape the phagocytosis of the mononuclear phagocyte system (MPS), and thus increases the distribution in tumor tissues and reduces liver toxicity. Conversely, when EVs overexpress SIRPα, due to binding to CD47 on the surface of cancer cells, EVs disrupt the interaction of the cancer cell signaling CD47-SIRPα axis, resulting in increased phagocytosis of cancer cells by macrophages, thereby suppressing tumor growth. Furthermore, SIRPα-EVs treatment promoted dense infiltration of T cells in a syngeneic cancer mouse model, raising the possibility that CD47-targeted therapy could unleash innate and adaptive antitumor responses [31]. In addition, it has been reported that CD24 can be a major innate immune checkpoint in ovarian and breast cancers and a promising target for cancer immunotherapy [70]. The authors demonstrate that tumor-expressed CD24 promotes immune evasion by interacting with the inhibitory receptor sialic acid-binding Ig-like lectin 10 (siglec-10) expressed by tumor-associated macrophages. In other words, CD24 can be modified like CD47, or as a complement to CD47, blocked in cancer cells expressing CD47 and CD24 simultaneously. Following the same line of thinking, candidate molecules such as MHC [71], PD-1/PD-L1 checkpoints [72,73], etc., can also be tested for their ability to make EVs invisible to the immune system, which will allow for greater modification flexibility and variety.

#### 3.3.2. Biochemical Strategies

Furthermore, as a means of chemical modification, PEGylated liposomes have been shown to limit opsonization and interact less with cells, thereby extending half-life in the bloodstream. Fusion of EVs with PEGylated liposomes can alter their membrane properties, thereby reducing their interaction with macrophages and prolonging circulation time [74]. In another research [75], an EV protein component, complement factor H (CFH), isolated from the plasma of lung adenocarcinoma patients and highly metastatic hepatocellular carcinoma (HCC), was also shown to be active in tumor cell-derived EVs and protect them from complement lysis and phagocytosis. In the future, more new mechanisms may be discovered to expand the “invisibility cloak” function of EVs.

### 3.4. To Improve Targeting Capability

Among these purposes, improving targeting ability has received the most attention, as stronger targeting ability can avoid side effects caused by drug retention in normal organs, and more aggregation at pathological sites can improve the therapeutic effect [39,98]. Next, we discuss EV modification strategies to improve organ targeting and molecular targeting.

#### 3.4.1. Organ Targeting

Due to their nanoscale size, EVs evade the vasculature through leaky endothelial tissue through a process known as the enhanced permeability and retention effect (EPR effect), having a natural advantage to aggregate at tumor sites for tumor therapy [99]. It has been demonstrated that EVs are intrinsically targeted, at least to some extent, because lipid composition and protein content can influence the tropism of EVs for specific receptors. For example, EVs from different human breast cancer cell sublines show tropism to different organs (brain, lung, or liver) due to their different surface integrin patterns, respectively [76,77].

Daniel J. Siegwart’s group reported a strategy called selective organ targeting (SORT) [100,101]. This strategy allows for the systematic engineering of nanoparticles based on different charges (cations, anions, and neutral ions) with accurate delivery to the lung, spleen, and liver of mice after intravenous injection. However, there is no similar report on EV modification. Combined with the previously reported fusion scheme of liposomes and EVs [65], the organ-targeting properties of liposomes could be used to selectively target hybrid EVs to different organs according to potential changes to achieve organ-targeting capabilities.

#### 3.4.2. Molecular Targeting

EVs have a natural tumor-homing effect probably due to the expression of tumor-targeting ligands or cell adhesion molecules [34,102]. Taking advantage of the intrinsic tumor-targeting feature of EVs, S.M. Kim et al. used tumor cell-derived EVs to deliver CRISPR/Cas9 plasmids targeting poly(ADP-ribose) polymerase-1 (PARP-1) for ovarian cancer therapy [103]. Furthermore, various ligands such as tumor-specific proteins and antibodies, peptides, and aptamers have been used to bind to specific surface receptors overexpressed on tumor cell membranes to improve the affinity of EVs to the tumor cell surface [81,83,104]. These ligands can be functionalized on EVs by genetic engineering or by post-modification. For example, the interleukin 3 receptor (IL3-R) is highly expressed in chronic myeloid leukemia (CML) cells, but low or absent in normal hematopoietic stem cells. D. Bellavia et al. [87] used genetic engineering technology to express the exosomal protein Lamp2b in HEK293T cells and fused it with interleukin 3 (IL3) fragment to achieve the effect of targeting CML cells in vitro and in vivo. Likewise, human epidermal growth factor receptor 2 (HER2) is highly expressed in a substantial proportion of breast, ovarian, and colon cancer cases. Therefore, gene pre-transfection of engineered ankyrin repeat proteins (DARPins), a specific ligand for HER2-positive cells, into parental cells to generate engineered EVs can achieve high HER2 binding affinity and specific tumor site targeting [78,79,80], which is a rapid and efficient method. Neuropilin-1 (NRP-1) is a transmembrane glycoprotein overexpressed in glioma cells and tumor vascular endothelium, but less or not expressed in normal nerve cells and other tissues. G. Jia et al. [105] linked RGERPPR peptide (RGE), which is a specific ligand of NPR-1 on the surface of EVs by copper-free click chemistry [106], facilitating fast and efficient post-chemical modification of EVs and the resulting glioma-targeting drug delivery.

## 4. Nucleic Acid Drugs Delivered by EVs for Cancer Therapy

All RNA-based therapeutics are large and/or highly charged macromolecules without the ability to cross lipid bilayers, ranging in size from 4–10 kDa for single-stranded antisense oligonucleotides (ASOs), to ~14 kDa for miRNAs and double-stranded siRNAs, to ~200 kDa for clustered regularly interspaced short palindromic repeats (CRISPR)- associated protein 9 (Cas9) single guide RNAs (sgRNAs), to 700–7000 kDa for self-replicating mRNAs [8]. The natural characteristics of EVs support their application in nucleic acid drug delivery systems, which can be loaded into EVs by various methods such as pre-transfection of cells, incubation, electroporation, extrusion, or sonication [92,107,108], depending on the properties of the various nucleic acids. We review the research and clinical progress of several common EV-delivered therapeutic nucleic acid cargoes (ASO, miRNA, siRNA, mRNA, and CRISPR/Cas9 system) in tumor therapy below, and several typical EV-loaded nucleic acid drug processes are shown in Figure 3.

### 4.1. EVs for Delivery of Small Nucleic Acid Drugs

Small nucleic acid drugs have received more and more attention because they can be flexibly encapsulated, co-encapsulated with other drugs, and have a wide range of applications [79,80,110].

#### 4.1.1. ASO

ASOs are DNA- or RNA-based highly specific nucleic acid polymers that result in gene silencing, steric hindrance, or alternative splicing for regulating the expression of target RNAs [111]. In addition, ASO is an effective means to inhibit endogenous miRNAs, and ASOs that silence miRNAs have been renamed as AMOs (anti-miRNA oligonucleotides). RNA-based ASOs can directly bind to mature strands of endogenous target miRNAs and block their function, and as potent miRNA inhibitors, ASOs have been widely used in cancer therapy [112].

Minh T.N. Le’s group previously reported the advantages of using EVs derived from red blood cells (RBCs) (RBCEVs) as a therapeutic vehicle because of its economical, readily available, and easily scalable, non-immunogenic, and non-carcinogenic properties. They demonstrated that ASO delivered using RBCEVs efficiently downregulated miR-125b and inhibited tumor growth in human breast cancer and AML xenograft mouse models via intratumoral and systemic administration, respectively [29]. In 2022, this group further reported that after multiple intratumoral injections of RBCEVs carrying 3p-125b-ASO, the RIG-I cascade pathway was activated, and high levels of type I interferons and immune cell infiltration were induced in the tumor microenvironment, and thus tumor cells were apoptotic [113]. Kim et al. [104] used T7 polypeptide-modified EVs to load the antisense oligonucleotide AMO-21 targeting miR-21 into EVs by electroporation for the treatment of glioma. AMO-21 using T7-Exo can effectively deliver AMO to the rat brain via tail vein injection and reduce the level of miR-21 in glioblastoma, thereby inducing the expression of programmed cell death protein 4 (PDCD4) and phosphatase and tensin homologs (PTEN) in the tumor, reducing the volume of the tumor.

Natural components of exosomal lipid bilayers including prostaglandin F2 receptor negative regulator (PTGFRN) have been shown to enhance the delivery of drug cargoes to myeloid cells such as TAMs. It was recently reported that PTGFRN^++^ EVs are undergoing clinical trials (ClinicalTrials.gov: NCT04592484) to support clinical translation of ASO (exoASO-STAT6) targeting transcription factors that control macrophage phenotype [114]. The researchers mixed PTGFRN^++^ EVs with a signal transducer and activator of transcription 6 (STAT6) ASO at a ratio of 1:1 at room temperature and incubated for 30 min. The purified exoASO-STAT6 significantly slowed the growth of tumors, and 50% of the mice tumors completely regressed. Efficient remodeling of the tumor microenvironment (TME) was also triggered, enabling TAM reprogramming, conversion of macrophages from M2 to M1 type, and generation of CD8 T cell-mediated responses.

#### 4.1.2. Single-Stranded miRNA

miRNAs are a class of highly conserved single-stranded RNAs with a length of 19–25 nucleotides, generally located in noncoding regions of the genome. Although they do not encode proteins, they play an important role in regulating gene expression [115,116]. The specific sequence of the 3′-untranslated region (3′-UTR) of miRNA is completely or partially complementary to the mRNA of its target gene, resulting in the degradation or translation inhibition of the target protein, thereby negatively regulating the target protein. Growing evidence suggests that miRNA-related gain or loss mutational processes may contribute to cancer development and progression [117].

Tumor suppressor (TS) miRNAs are an attractive target for tumor therapy. Although cell-secreted EVs have endogenous miRNAs that can fight cancer [5,40,41,98], we want to discuss here that EVs serve as carriers to deliver additional encapsulated miRNAs, highlighting the role of EV carriers. For example, let-7a miRNA acts as a tumor suppressor and inhibits the malignant growth of cancer cells by reducing the expression of RAS and HMGA2. Ohno et al. [107] used HEK293-derived EVs overexpressing the GE11 peptide to efficiently deliver let-7a miRNA to epidermal growth factor receptor (EGFR)-expressing breast cancer cells. Let-7a inhibits breast cancer growth by altering cell cycle progression and reducing cell division. Similarly, Kobayashi et al. [118] purified EVs from primary cultured ovarian cancer patient omental fibroblasts, loading miR-199a-3p into EVs by electroporation, and the expression level of intracellular miR-199a-3p was increased by thousands to tens of thousands of times in different ovarian cancer cells. miR-199a-3p-Exo inhibited the expression of the direct target c-Met, thereby inhibiting cell proliferation and invasion. After treatment, the peritoneal spread of the ovarian cancer mouse model was significantly inhibited, and extracellular regulated protein kinases (ERK) phosphorylation and matrix metallopeptidase 2 (MMP2) expression in tumors were reduced.

#### 4.1.3. Double-Stranded siRNA

siRNAs are short (20–25 nucleotides) double-stranded RNA molecules, when delivered into the cytoplasm, argonaute 2 (AGO2) cleaves the passenger (sense) strand, while the guide (antisense) strand of the siRNA is loaded into the RNA-induced silencing complex (RISC). Then, the guide strand directs RISC to the recognized target mRNA for cleavage. RISC and guide strands can be recycled, so one siRNA molecule can drive the cleavage of multiple mRNA molecules for an efficient gene silencing [119]. Since siRNA is artificially synthesized, an accurate, and effective sequence design for the target is a critical step.

Although commercial liposomes have been around for a long time, using PANC-1 cells-derived EVs as a transfection vehicle resulted in higher knockdown efficiency compared to Lipofectamine™ transfecting P21-activated kinase 4 (PAK4) siRNA (siPAK4) at the same dose of siPAK4 (30 nM) (*p* < 0.01) [42]. This may be because the delivery mode of EVs is more flexible than liposomes since natural sources are more in line with the biological process of cells. Compared with transfection reagents, which have potential chemical transfection toxicity and may not be suitable for clinical treatment, the development of EV transfection reagents has a wide range of uses and a huge market.

By encapsulating si-survivin by electroporation, MSC EVs expressing CXC chemokine receptor type 4 (CXCR4) can specifically bind to the highly expressed stromal cell-derived factor-1 (SDF-1) on the tumor surface, reach the tumor site, knock down the survivin gene, and achieve a tumor-killing effect [120]. KRAS^G12D^, the most common oncogene KRAS mutation, has received extensive attention as a promising target for the treatment of solid tumors. CD47-functionalized MSC EVs carrying KRAS^G12D^ siRNA are a promising vehicle for reducing KRAS expression in patient-derived xenograft mice, resulting in cancer cell apoptosis, inhibition of metastasis, and increased overall survival without cytotoxic effect [67]. Good Manufacturing Practice (GMP) certified MSC EV production has been approved by the FDA (Food and Drug Administration) for clinical trials, and this strategy is currently undergoing clinical evaluation in the Phase I trial (NCT03608631) [121].

#### 4.1.4. Combination Therapy

Recent studies have shown that RNA and chemotherapeutic drugs can be co-encapsulated in engineered EVs to sensitize gene drugs or act synergistically to achieve better antitumor effects. Gong C et al. [122] pre-transfected a disintegrin and metalloproteinase 15 (A15) with THP-1 cells to obtain A15-Exo, targeting tumor cells highly expressing integrin α_v_β_3_, and doxorubicin (DOX) was then incubated overnight in triethylamine solution, and next cholesterol-modified miR-159 was incubated with shaking at 37 °C to form co-encapsulated A15-Exo/Cho-miR-159. The results showed that the antitumor effect of the combination drug was significantly stronger than that of the single drug, and the presence of A15 on the surface of EVs played a decisive role in the delivery of the drug to the tumor. In another study [123], researchers first loaded siRNA into engineered lipid hybrid EVs (eEVs) by electroporation. Subsequently, the chemotherapeutic drug DOX was incorporated into the polyelectrolyte shell around the eEVs, which was deposited by the layer-by-layer assembly (LBL). The LBL-eEV complex can be transported through cells and release siRNA in the cytoplasm, while the delivered DOX enters the nucleus to induce programmed cell death. The inherent selectivity of LBL-eEVs for cancer cells resulted in efficient gene silencing and cancer-killing rates while reducing cytotoxicity to normal cells. Zhou et al. [109] designed a delivery system, which selected bone marrow mesenchymal stem cells (BM-MSCs) as EV-derived cells, loaded with Galectin-9 siRNA by electroporation, and surface-modified oxaliplatin (OXA) prodrug by vortexing as immunogenic cell death (ICD)-trigger. This combination therapy (iEXO-OXA) induces antitumor immunity through tumor suppressor macrophage polarization, cytotoxic T lymphocyte recruitment, and Tregs downregulation, resulting in significant efficacy in cancer therapy.

In addition, co-delivery also has a better therapeutic effect on drug-resistant tumors [124]. Li L et al. [82] fused CD47-expressing tumor EVs with cRGD-modified liposomes for co-delivery of miR-497 and triptolide. In vitro experimental results showed that nanoparticles could be effectively taken up by tumor cells, thereby significantly promoting tumor cell apoptosis, promoting the production of reactive oxygen species (ROS), and upregulating the polarization of macrophages from M2 to M1. Mechanistically, they promote dephosphorylation of the overactivated phosphatidylinositol 3-kinase/protein kinase B/mammalian target of rapamycin (PI3K/AKT/mTOR) signaling pathway, reverse cisplatin resistance in ovarian cancer, and exert antitumor effects. Pranela Rameshwar’s group earlier explored if the miR-9 expression was elevated in temozolomide (TMZ)-resistant glioblastoma (GBM) cells. Anti-miR-9 can block and downregulate miR-9, mediates the downregulation of drug-resistant P-gp expression, and reverse the expression of multidrug transporters to sensitize GBM cells to TMZ. The combined application of anti-miR-9 enhanced cell death compared to TMZ alone [125]. These data demonstrate that optimization studies on RNA regulation in combination with chemotherapeutic agents may help to establish synergistic treatments that may soon be translated into the clinic.

For antitumor immunity, the co-loading of Toll-like receptor 3 (TLR3) agonist Hiltonol, a double-stranded RNA that also acts as an immune adjuvant, and the ICD inducer neutrophil elastase (ELANE) into EVs expressing α-lactalbumin (α-LA) by electroporation (HELA-Exos), can be designed as an antitumor vaccine for breast cancer treatment to achieve in situ activation of DCs. The results showed that HELA-Exos significantly inhibited tumor growth, enhanced the antigen cross-presentation of DCs, and generated antitumor CD8^+^ T cells in a patient-derived tumor–organ co-culture system. This provides a direct reference for EV delivery of tumor nucleic acid vaccines [126].

### 4.2. EVs for Delivery of mRNA

mRNA-based therapy is the delivery of artificially synthesized mRNA to specific cells that encodes therapeutically active proteins in the cytoplasm [127]. The concept of introducing mRNA directly into cells instead of DNA has been around for decades. For therapeutic purposes such as protein replacement therapy, messenger RNA is considered a safer alternative to DNA because it can be rapidly degraded without fear of potential adverse effects of long-term expression or genome integration [128].

Notably, Yang et al. [52] reported that CNP enables large-scale generation of functional mRNA-encapsulated EVs for targeted transcriptional manipulation for glioma therapy. They developed a CNP system to stimulate cells to produce and release mRNA-containing EVs. Nanochannels (approximately 500 nm in diameter) can shuttle DNA plasmids containing PTEN and CDX (CD47 cloning targeted peptide) from buffers into attached cells by transient electrical pulses. The extent of EV secretion can be controlled by adjusting the voltage of the nanochannel, as the voltage increased from 100 V to 150 V, the number of released EVs gradually increased until a plateau was reached at 200 V, and the loading of PTEN mRNA is nearly 1000 times that of Lipo2000 and ordinary electroporation. After the transfer of PTEN mRNA into glioma cells based on CNP technology, the deletion of PTEN was changed and the expression of the tumor suppressor gene PTEN protein was upregulated, thus playing a tumor suppressor role.

In addition to expressing tumor suppressor genes themselves, mRNA can also catalyze prodrugs and play an antitumor effect, that is, gene-delivered prodrug therapies. Engineered MSC-derived EVs provide an alternative tool to deliver suicide gene mRNA for targeted cancer gene therapy. According to Altanerova et al. [129], the application of dental pulp MSC-derived EV-transduced yCD::UPRT therapeutic mRNA significantly inhibited tumor growth in glioblastoma. In the next study, they reported the catalysis of prodrugs by EVs with yCD::UPRT released from MSC. The tumor-killing mechanism of yCD::UPRT therapeutic mRNA depends on the conversion from nontoxic 5-FC to highly cytotoxic 5-FU [130]. Similarly, A.C. Matin’s group constructed an engineered EV capable of targeting HER2-positive breast cancer cells, encapsulating HChrR6 mRNA and prodrug CNOB (C_16_H_7_CIN_2_O_4_) to reach tumor cells, using HChrR6 to convert the CNOB activation into the cytotoxic drug MCHB (C_16_H_9_CIN_2_O_2_) [131,132].

Judging from the current trend, the research on mRNA vaccines is very hot, and many companies have invested in the research and development of LNP-mRNA vaccines. The basic principle of the mRNA vaccine is to introduce the mRNA expressing the antigen target into the body through a specific delivery system, express the protein in the body, and stimulate the body to produce a specific immunological response, so that the body can obtain immune protection [133]. However, some studies have pointed out that after LNP delivers mRNA, EVs of cells will secrete some mRNAs outside the cell. On the one hand, this process prevents the formation of proteins that should not be produced outside the cell and reduces damage to other parts. On the other hand, it also shows that more research is needed to determine how much LNP delivery is achieved by the LNP’s contribution rather than Endo-EV from individuals receiving LNP therapy [134]. Recently, it has been reported that the use of Gram-negative bacteria-derived outer-membrane vesicles (OMVs) as an mRNA delivery platform strongly stimulates the innate immune system, promoting antigen presentation and T cell activation. Moreover, OMVs generated after surface modification of the RNA-binding protein L7Ae and the lysosomal escape protein Listerinolysin O (OMV-LL-mRNA) can significantly inhibit tumor progression, induce long-term immune memory, and protect mice from tumor challenges after 60 days [135].

### 4.3. EVs for Delivery of CRISPR/Cas9 System

Genome editing technology has emerged as a potential tool for the treatment of incurable and rare diseases [136]. In particular, the discovery of CRISPR/Cas9 systems and the design of sgRNAs have further brought the development of RNA therapeutics to the forefront [137]. Co-delivery of Cas9 and sgRNA targeting a genomic target holds promise for gene knockout strategies [9,138]. Researchers have demonstrated the possibility of using the CRISPR/Cas9 system to treat various diseases by repairing, deleting, or silencing certain genetic mutations associated with the disease in the body [139,140,141]. The following describes the delivery of CRISPR systems in the form of plasmids, RNAs, and ribonucleoprotein (RNP) using EV as a carrier.

#### 4.3.1. Plasmids

Compared with EVs derived from HEK293 cells, EVs derived from SKOV3 cells showed enhanced cellular uptake of SKOV3 receptors. Due to the tropism of homologous cells, Kim et al. [103] introduced expression plasmids carrying Cas9 and poly (ADP-ribose) polymerase-1 (PARP-1) sgRNAs into SKOV3 ovarian cancer cells by electroporation. Intravenous injection of SKOV3-derived EVs showed significant accumulation in tumor tissues of SKOV3 xenograft mice and resulted in significant PARP-1 knockout, exhibiting 27% indel efficacy. Since PARP-1 is primarily involved in the DNA repair process, the authors of this study combined EV therapy with cisplatin, which induces DNA damage, and found that decreased expression of PARP-1 enhanced the chemosensitivity of cancer cells to cisplatin, showing synergistic cytotoxicity. The cancer proliferation inhibition rate of this combined antitumor regimen was 57%, while that of the EV system and cisplatin alone were 30% and 21.6%, respectively. Additionally, to enhance the loading of genome editing components in EVs, Lin et al. [65] designed a strategy to fuse EVs and liposomes for plasmid loading. The researchers incubated EVs derived from HEK293FT cells with plasmid DNA-liposome complexes. Subsequent fusion of EVs and liposomes enables plasmid DNA to be encapsulated in hybrid EVs. Hybrid EVs could significantly enhance the expression of plasmid DNA encoding enhanced green fluorescent protein (EGFP), and 13.2% of EGFP-positive cells were obtained, compared with 0.11% and 1.46% of EGFP-positive cells in the EV-only group and liposome-treated group, respectively. This provides an idea for the fusion EV packaging and delivery CRISPR/Cas9 system [142].

#### 4.3.2. sgRNA and Cas9 mRNA

For the previously introduced RBCEV system, the researchers also attempted to electroporate Cas9 mRNA together with 125b-gRNA into RBCEVs [29]. After 2 days of treatment of MOLM13 cells with this system, different insertions and deletions were found at the cleavage site, disrupting the mature miR-125 sequence, resulting in a 98% reduction in miR-125b expression and a 90% reduction in miR-125a expression. Meanwhile, RBCEV protected electroporated Cas9 mRNA from RNase I_f_-mediated degradation, and approximately 18% of the Cas9 mRNA was loaded and protected in RBCEV. In addition, the CD9-HuR functionalized EVs constructed by Li et al. [66] introduced in the previous chapter have a strong ability to enrich specific RNAs. These functionalized EVs were used to deliver deactivated Cas9 (dCas9) mRNA to the target gene CCAAT enhancer binding protein α (C/ebpα) gRNA associated with liver cell proliferation and differentiation. Compared with free dCas9 mRNA, the expression of the target gene was reduced by approximately 20-fold.

#### 4.3.3. RNP

The reasons for the gradual acceptance of RNP delivery are: (1) it is easier to detect the activity of Cas9 protein than Cas9 mRNA, (2) it has a shorter duration of action, and (3) RNP can be degraded as quickly as possible in target cells, and this can reduce unnecessary off-target effects [143]. Based on this, Yao et al. [144] used the loading method of the RNA fusion protein to insert the RNA aptamer Com into sgRNA and fused Com to both ends of CD63. Com/com interaction enriches Cas9 and adenine base editor (ABE) RNPs into EVs by forming a three-component complex including CD63-Com fusion protein, Com-modified sgRNA, and Cas9 or ABE. Thus, the genome editing and transient expression capabilities of RNP-enriched EVs are effectively achieved. However, the approach adopted by Zhuang et al. [145] is relatively simple: they incubated Cas9 protein and sgRNA at 37 °C for 10 min to form the RNP complex, and then sonicated or repeated freeze–thaw of the RNP, and repeated freeze–thaw loading was found to be more efficient. Next, valence-controlled tetrahedral DNA nanostructures (TDN) were surface-modified and TDN was loaded onto the surface of EV via cholesterol anchoring for specific cellular targeting. The system is ultimately used to reduce the protein expression of WNT10B, which has a significant inhibitory effect on tumor growth in vitro and in vivo and can be extended to other therapeutic targets.

Delivering CRISPR therapy is challenging because Cas proteins and sgRNAs must be present in sufficient concentrations to form intracellular ribonucleoproteins [127]. An ideal genome-editing nuclease delivery system should overcome various hurdles to achieve precisely targeted delivery while protecting the genome-editing nucleases until they reach target tissues or cells [146]. Once the long-term safety and clinically relevant issues of CRISPR delivery are addressed, products of this powerful tumor therapy tool can be rapidly rolled out, and these issues can be addressed incrementally with EVs. Table 2 lists the EV delivery strategies for nucleic acid drugs.

### 4.4. Clinical Status of EV-Based Nucleic Acid Drug Delivery Systems

At the same time, we can also see that the clinical transformation of nucleic acid drugs based on EVs as carriers is on the rise (Table 3). There are already many clinical trials and upcoming drugs for the development of nucleic acid drugs for EV delivery [151]. Codiak BioSciences is the world’s first public biopharmaceutical company focused on developing EV therapeutics. Codiak currently develops three clinical pipelines: exoSTING, exoIL12, and exoASO-STAT6 [114], which represent EVs that deliver small molecules, protein macromolecules, and nucleic acid-based drugs into the body, respectively. At present, exoSTING and exoIL12 have completed phase I clinical and are in the phase II clinical stage, and the safety data of these two pipelines are performing well, while exoASO-STAT6 has just entered phase I clinical stage [152]. EVOX Therapeutics, a British company, develops products loaded with mRNA and siRNA EVs through its self-developed DeliverEX^TM^ platform to make up for the lack of metabolism-related enzymes, and is expected to make breakthroughs in the treatment of rare diseases and neurological diseases [153]. Better efficacy or better-targeted delivery provided by this delivery technology could mean lower doses of the drug can be given and potentially lower costs [154].

China’s EV industry is also booming. Echo Biotech [155] has established a complete production process and quality control system for medicinal EVs and is about to complete the construction of the first GMP pilot plant for engineered EVs in China. The engineered EVs technology platform Echosome^®^ can realize the efficient carrying and delivery of different types of drugs (proteins, nucleic acids, small molecules), and adopt endogenous engineering (pre-modification) and exogenous engineering (post-modification) solutions depending on the type of molecule to improve the utilization of protein and small molecule drugs and the intracellular delivery of functional proteins and targeted delivery of nucleic acid drugs, etc. VesiCURE [156] independently developed modEXO^TM^, an engineered EV drug delivery platform, and selected several main directions for EVs to treat tumors, liver diseases, lung diseases, and digestive system diseases, and hope to achieve precise targeted clinical needs in the future.

## 5. Conclusions and Research Prospects

Recent advances in research provide hope for the treatment of many rare or incurable diseases, including cancer. We reviewed the problems encountered by EVs in industrial applications as well as the progress of scientific research in this article, which served as a reference for the future large-scale development of EV-based nucleic acid drug systems. At the same time, we also found that EV delivery still faces some problems.

**Quality control is a problem**. Large-scale batch production and purification of EVs have not yet formed a unified process, which may make different batches of EVs different [157]. In terms of product quality inspection, for example, EV’s quantity, size, surface marker expression, microbial contamination, and specific functional activity, there are also no recognized indicators for product quality assessment [16].

**The drug loading rate is still not ideal**. Whether the parental cells are manipulated for pre-separation and encapsulation or encapsulation after EV extraction, the drug encapsulation rate remains relatively low, which cannot reach the encapsulation rate and drug load that the drug delivery carrier should have, limiting its clinical application [158].

**Stability is difficult to guarantee**. To achieve homogeneity and stability of EVs, the long-term storage of EVs is also a major problem [159]. To protect its biological activity and facilitate transportation and clinical application, freeze-drying and spray-drying can maximize the shelf life of EVs, but the cost is expensive and the storage period is still short [160].

**Metabolism and dynamics tracking are difficult to achieve**. During the in vivo drug delivery process of EVs, on the one hand, the contents are continuously decomposed and released, and on the other hand, EVs are adsorbing different molecules and always changing dynamically. As a result, accurate prediction, monitoring, and control of the biodistribution of EVs are critical to their successful development as drug-delivery vehicles and therapeutics. Biological imaging, fluorescence imaging, and nuclear imaging are currently being used in EV tracking. Considering both temporal and spatial resolution, SPECT/CT provides very accurate biodistribution and anatomical localization of EVs; however, these techniques have not been widely used due to the need for hazardous radioisotopes, specialized infrastructure, and equipment [161].

In the future, commercial EV production requires stricter GMP specifications, including the selection of EV sources, standardized cell culture techniques, downstream EV isolation, purification, and quality assessment protocols, EV detection and tracking, and the stability of EV-based preparations [162].

## Figures and Tables

**Figure 1 pharmaceutics-14-02236-f001:**
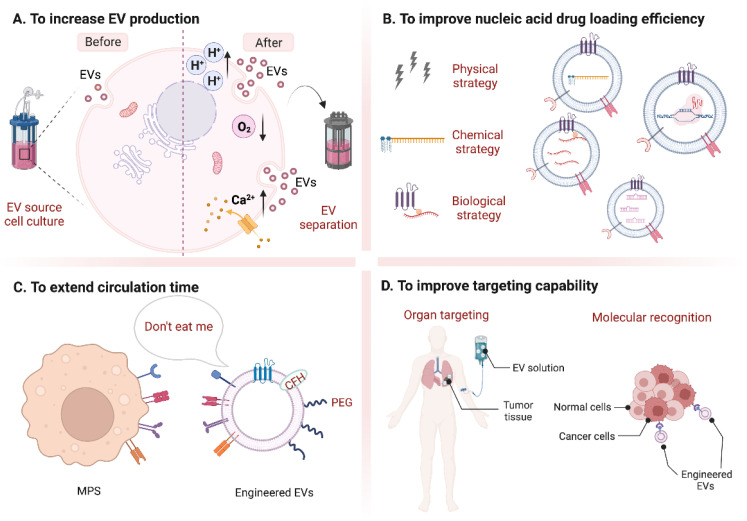
Summary of different modification strategies for EVs. (**A**) to increase EV production; (**B**) to improve nucleic acid drug loading efficiency; (**C**) to extend circulation time; (**D**) to improve targeting capability. (Created with BioRender.com, accessed on 23 August 2022).

**Figure 2 pharmaceutics-14-02236-f002:**
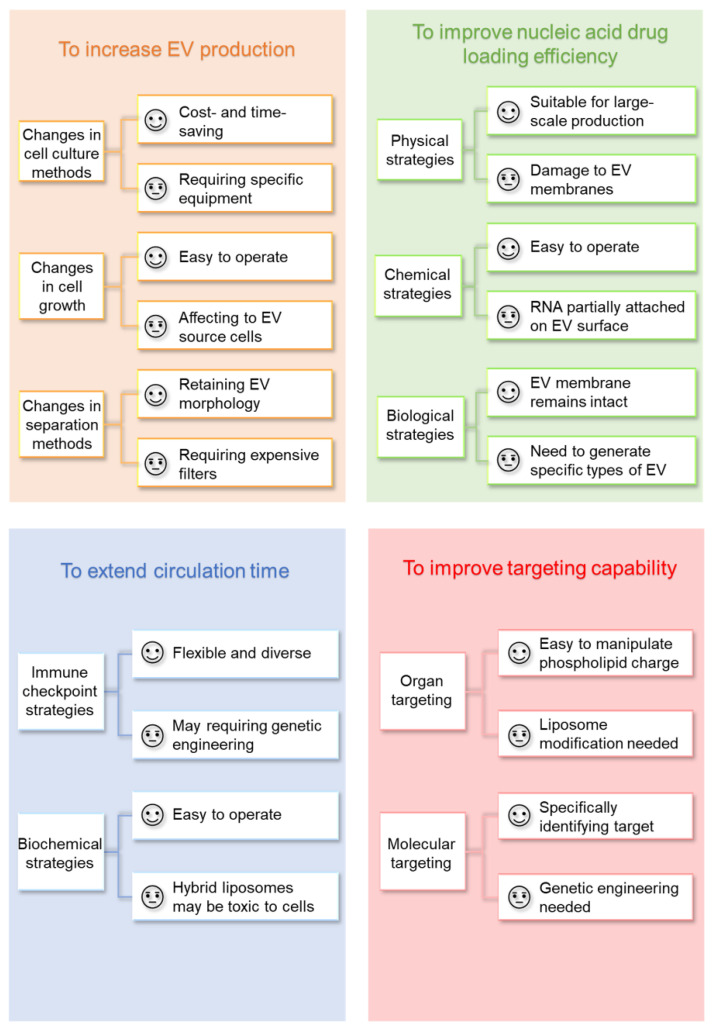
Comparison of various EV modification strategies.

**Figure 3 pharmaceutics-14-02236-f003:**
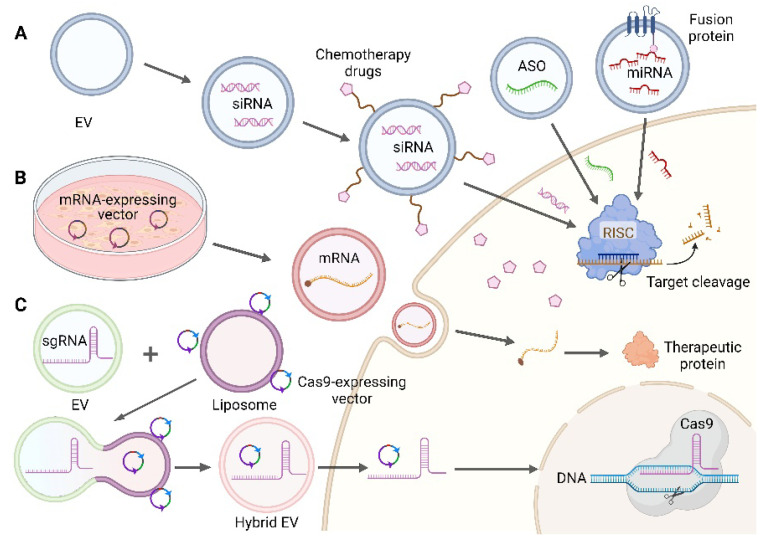
Several typical methods of EV loading nucleic acid. (**A**) Loading of small nucleic acid drugs: ASO, using fusion protein method to load miRNA, co-encapsulate siRNA, and chemotherapeutic drugs. Adapted with permission from ref. [66,109]. Copyright 2019, American Chemical Society and Copyright 2020, Elsevier B.V. (**B**) Loading of mRNA: EV-derived cells were pre-transfected to express mRNA plasmid. (**C**) Loading of CRISPR/Cas9: EVs loaded with sgRNA were fused with liposomes loaded with Cas9-expressing vectors to form hybrid EVs. Adapted with permission from ref. [65]. Copyright 2018, Joh Wiley and Sons. (Created with BioRender.com, accessed on 24 August 2022).

**Table 1 pharmaceutics-14-02236-t001:** A summary of EV modification strategies.

Purposes	Strategies	References
To increase EV production	Changes in cell culture methods (specially made culture flasks or 3D culture)	[42,43,44,45,46,47]
	Ca^2+^-dependent regulation induction	[48,49,50,51]
	Stressed culture conditions stimulation (hypoxia, low pH, electrical stimulation, liposome stimulation, and drug stimulation)	[52,53,54,55,56,57]
	Changes in EV separation methods (tangential flow filtration)	[58,59,60]
To improve nucleic acid drug loading efficiency	Physical strategies (1. cellular nanoporation biochips; 2. pH gradient; 3. extrusion)	[38,52,61,62]
	Chemical strategies (hydrophobic modification of nucleic acids)	[63,64]
	Biological strategies (fusion with liposomes and RNA fusion proteins)	[65,66]
To extend circulation time	Immune checkpoint strategies (CD47, CD24, MHC, PD-1/PD-L1)	[31,67,68,69,70,71,72,73]
	Biochemical strategies (PEGylation and complement factor H)	[74,75]
To improve targeting capability	Organ targeting	[76,77]
	Molecular targeting (EGFR, A33, CD44, HER2, et al.)	[78,79,80,81,82,83,84]

**Table 2 pharmaceutics-14-02236-t002:** A summary of various nucleic acid drug strategies encapsulated by EVs.

Loading Strategies	Loaded Drugs	Notable Details	References
Electroporation	ASO	Target genes can be inhibited by up to 95% at the mRNA level.	[29]
	siRNA	The drug loading rate is close to 60%.	[41]
	ASO	Upregulation of tumor suppressor genes and tumor suppressors.	[42]
	CRISPR/Cas9	Electroporation of Cas9-/sgRNA-expressing plasmids into EVs.	[61,62,63,64]
	siRNA	EV transfects siRNA better than commercial transfection reagents Lipofectamine^TM^ and polyethyleneimine (PEI).	[94]
	siRNA	EVs are hybridized with liposomes through membrane extrusion technology.	[103]
Incubation	siRNA	EVs were briefly incubated with siRNA in the presence of PEI.	[94]
	ASO	ASO is attached to the membrane of EVs.	[114]
	miRNA and DOX	EVs were first mixed with DOX and then incubated with miRNA with shaking.	[122]
Pre-transfected source cells	mRNA	Introducing plasmids into cells to obtain mRNA-carrying EVs.	[131,132]
Repeated freeze–thaw cycles	CRISPR/Cas9	Cas9 protein was incubated with sgRNA to form RNP	[145]
Using transfection reagents	siRNA and miRNA	Using Exo-Fect^TM^ Exosome Transfection Kit	[147]
	CRISPR/Cas9	Transfection of encoding plasmids using the Exo-Fect^TM^ Exosome Transfection Kit.	[148]
	miRNA	Using Lipofectamine^TM^ RNAiMAX Reagent Kit	[149]
	miRNA	Using ExoFectin^®^ Kit	[150]

**Table 3 pharmaceutics-14-02236-t003:** A summary of clinical transformation of nucleic acid drugs based on EVs for cancer therapy.

Country	Institution	Research Content	Payload	Clinical Trial Number
United States	Codiak BioSciences	exoASO^TM^-STAT6	ASO	NCT05375604 (Phase I)
United States	M.D. Anderson Cancer Center	iExosomes	KRAS^G12D^ siRNA	NCT03608631 (Phase I)
United States	Vesigen Therapeutics	ARMMs technology	RNA/Gene editing	/
United States	Carmine Therapeutics	REGENT^®^ platform	RNA/Gene editing	/
England	EVOX Therapeutics	DeliverEX^TM^ platform	mRNA/Gene editing	/
China	Echo Biotech	Echosome^®^ platform	/	/
China	VesiCURE	modEXO^TM^	/	/
Korea	EXOSOME plus	ExoThera^TM^	/	/
Switzerland	Anjarium Biosciences	Hybridosome^®^ delivery technology	DNA	/

## Data Availability

Not applicable.

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
