# Peer review of "Extracellular Vesicles as Delivery Vehicles for Therapeutic Nucleic Acids in Cancer Gene Therapy: Progress and Challenges"

_pharmaceutics, 2022, doi:10.3390/pharmaceutics14102236_

Round 1
Reviewer 1 Report
The manuscript entitled ‘Extracellular Vesicles as Delivery Vehicle for Nucleic Acids: Progress and Challenges’’ by Du et al. described that EV-loaded nucleic acids can change gene expressions and functions of recipient cells including RNA-based nucleic acid drugs and CRISPR/Cas gene-editing systems. They discuss the techniques and methods to increase EV yield, enhance nucleic acid loading efficiency, extend circulation time, and improve targeted delivery, as well as their applications in gene therapy and combination with other tumor therapiesI think these results are important in field, however, I provided comments on the manuscript that authors should address them accordingly.
1. Authors correct typo errors and English grammar throughout the text.
2. Improve the abstract by presenting findings.
3. Exosomes are 30-150 nm extracellular vesicles, therforte revise it and uniform it thruouth the text.
4. Authors should cite references for exosomes biogenesis and roles in cancer (PMID: 34890589). These references are useful for many sections of this manuscript.
5. Uniform EVs or EV.
6. Authors added a brief (1-2 line) about ISEV.
7. Authors uniform citation throught the text like ‘’ communication [35]. (4)’’.
8. Remove or expand the section ‘’Identifying specified biomarkers’’ this section is too simple.
9. For combanition therapy and engineering exosomes use this reference: PMID: 36075472.
10. Increase Fig. 2 resoulation.
Author Response
We are very grateful to Reviewer for reviewing the paper so carefully. Based on your comments, we have made some changes, and the point-by-point responses are as follows:
Point 1: Authors correct typo errors and English grammar throughout the text.
Response 1: We have corrected spelling errors and English grammar throughout the text, with changes marked in red.
Point 2: Improve the abstract by presenting findings.
Response 2: Many studies have demonstrated that EV-loaded nucleic acids, including RNA-based nucleic acid drugs and CRISPR/Cas gene-editing systems, can alter gene expressions and functions of recipient cells for cancer gene therapy. Here in this review, we discuss the advantages and challenges of EV-based nucleic acids delivery systems in cancer therapy. We summarize the techniques and methods to increase EV yield, enhance nucleic acid loading efficiency, extend circulation time, and improve targeted delivery, as well as their applications in gene therapy and combination with other cancer therapies. Finally, we discuss the current status, challenges, and prospects of EVs as a therapeutic tool for the clinical application of nucleic acid drugs.
Point 3: Exosomes are 30-150 nm extracellular vesicles, therforte revise it and uniform it thruouth the text.
Response 3: We discuss extracellular vesicles for nucleic acids delivery in this review. To clarify the difference between extracellular vesicles and exosomes, we add the definition of extracellular vesicles and exosomes on page 2 line 69-76, and the works we cite that focus on exosomes are also point out in the revised manuscript. Modification on page 2 line 68-77: They can be endosome-derived (termed exosomes, diameter 30 ~ 150 nm) or are generated by membrane outward budding (termed ectosomes, diameter 50 ~ 1000 nm) [15]. From either biogenesis, EVs are natural vehicles carrying and transferring biological information for cellular communication and have attracted increasing attention as drug delivery systems.
Point 4: Authors should cite references for exosomes biogenesis and roles in cancer (PMID:34890589). These references are useful for many sections of this manuscript.
Response 4: According to your kind comment, we have cited this paper, which is at [17] in revised manuscript.
Point 5: Uniform EVs or EV.
Response 5: We uniform the abbreviation as EVs throughout the text. Changes are marked in red throughout the text.
Point 6: Authors added a brief (1-2 line) about ISEV.
Response 6: We have added a brief introduction to ISEV at line 63 to line 65 in revised manuscript as follows: The International Society for Extracellular Vesicles (ISEV) is a professional social group composed of researchers and scientists in the field of EVs. It is committed to promoting global EV research and is one of the most authoritative societies in the field of EVs.
Point 7: Authors uniform citation throught the text like ‘’ communication [35]. (4)’’.
Response 7: We are very sorry for the misunderstanding. We uniformly use square brackets as the literature citation format in the full text, such as "[35]".
"(4)" is the serial number of “However, as the drug delivery system, the following aspects still need to be considered:” in the previous text. (Page 2, line 98)
Point 8: Remove or expand the section ‘’Identifying specified biomarkers’’ this section is too simple.
Response 8:
According to the reviewer, we expand this section in the revised manuscript as follows:
EVs have natural tumor-homing effect probably due to the expression of tu-mor-targeting ligands or cell adhesion molecules [34,102] Taking advantage of the in-trinsic tumor-targeting feature of EVs, S.M. Kim et al. used tumor cell-derived EVs to deliver CRISPR/Cas9 plasmids targeting poly(ADP-ribose) polymerase-1 (PARP-1) for ovarian cancer therapy [103]. Furthermore, various ligands such as tumor-specific pro-teins and antibodies, peptides, aptamers have been used to bind to specific surface re-ceptors overexpressed on tumor cell membranes to improve the affinity of EVs to the tumor cell surface [82,84,104]. These ligands can be functionalized on EVs by genetical engineering or by post-modification. For example, the interleukin 3 receptor (IL3-R) is highly expressed in chronic myeloid leukemia (CML) cells, but low or absent in normal hematopoietic stem cells. D. Bellavia et al. [88] used genetic engineering technology to express the exosomal protein Lamp2b in HEK293T cells and fused it with interleukin 3 (IL3) fragment to achieve the effect of targeting CML cells in vitro and in vivo. Likewise, human epidermal growth factor receptor 2 (HER2) is highly expressed in a substantial proportion of breast, ovarian, and colon cancer cases. Therefore, gene pre-transfection of engineered ankyrin repeat proteins (DARPins), a specific ligand for HER2-positive cells, into parental cells to generate engineered EVs can achieve high HER2 binding affinity and specific tumor site targeting [79-81]. which is a rapid and efficient method. Neuro-pilin-1 (NRP-1) is a transmembrane glycoprotein overexpressed in glioma cells and tumor vascular endothelium, but less or not expressed in normal nerve cells and other tissues. G. Jia et al. [105] linked RGERPPR peptide (RGE) that is a specific ligand of NPR-1 on the surface of EVs by copper-free click chemistry [106], facilitating fast and efficient post-chemical modification of EVs and the resulting glioma-targeting drug delivery.
Point 9: For combanition therapy and engineering exosomes use this reference: PMID: 36075472.
Response 9: According to your kind comment, we have cited this paper, which is at [124] in revised manuscript.
Point 10: Increase Fig. 2 resoulation.
Response 10: We have increased the resolution of Fig.2 to 600 dpi.
Reviewer 2 Report
Overall, Extracellular Vesicles as Delivery Vehicle for Nucleic Acids: Progress and Challenges is an excellent review. The authors have done a great job in handling the topic with proper flow. I would like to thank the authors for their strong, clear, and organized writing.
Please address the following comments.
Comment 1: The following two subheadings have the same name.
3.1.1. Changing cell culture methods.
3.1.2. Changing cell culture methods.
Comment 2: In figure captions, please replace “Accepted with permission from ref” with reproduced/adapted with permission from ref.
Cite the following latest references
https://doi.org/10.3390%2Fpharmaceutics12100980
https://doi.org/10.3390/nano11061481
https://doi.org/10.1016/j.addr.2021.03.005
Author Response
Thank you for your valuable comments that help improve the quality of our review papers. Based on your comments, we have made appropriate revisions, and the point-by-point responses are as follows:
Comment 1: The following two subheadings have the same name.
3.1.1. Changing cell culture methods.
3.1.2. Changing cell culture methods.
Response to Comment 1: We're sorry for the typo, and thank you for pointing it out. The subheading 3.1.2. has been changed to "Changing EVs separation methods". (Page 5, line 160)
Comment 2: In figure captions, please replace “Accepted with permission from ref” with reproduced/adapted with permission from ref.
Cite the following latest references
https://doi.org/10.3390%2Fpharmaceutics12100980
https://doi.org/10.3390/nano11061481
https://doi.org/10.1016/j.addr.2021.03.005
Response to Comment 2: Thank you for your precise request. We have replaced “Accepted with permission from ref” with "Adapted with permission from ref" based on your comments. (Page 11, line 424 and line 427)
At the same time, according to your request, we have added citations of three papers, which are at [36] [159] [142] in revised manuscript.
Reviewer 3 Report
In the present manuscript entitled: “Extracellular vesicles as delivery vehicle for nucleic acids: progress and challenges” by Du et al, the authors reported and further discussed the techniques and methodologies to increase extracellular vesicles yield, improve targeting delivery, their applications mainly for cancer therapies.
The introduction is updated and complete according to the aim of the present manuscript. Although, I suggest including the specific approach for cancer treatments/therapy in the title.
Moreover, the manuscript is organized and focused on the topic that is of increasing importance due to the prospective biological purposes for cancer treatments.
Besides, it is important to consider possible corrections/additions/suggestions to improve the manuscript for the readers:
- Addition of one table/image: Comparison as a summary (which are the advantages/disadvantages?) among the methodologies regarding to increase EV production, to improve nuclei acid drug loading efficiency, to extend circulation time, and to improve targeting capability
- Suggestion. Item: Clinical status of EV-based nucleic acid drug delivery systems: include one table summarizing all the information
- Some additional references to consider:
1) Which is the additional information the present review article has in comparison to Pharmaceutics. 2020; 12(10): 980. doi: 10.3390/pharmaceutics12100980. Extracellular Vesicle-Based Nucleic Acid Delivery: Current Advances and Future Perspectives in Cancer Therapeutic Strategies?
2) Important to add the following reference: Extracell Vesicles Circ Nucleic Acids 2022; 3:14-30. doi: 10.20517/evcna.2021.21. Nucleic acid functionalized extracellular vesicles as promising therapeutic systems for nanomedicine
Additionally, I would like to note the complete and interesting conclusions wrote topic by topic as an excellent summary.
Finally, I would like to invite the authors to add the abbreviation list of words at the end of this manuscript.
I recommend the acceptance of this manuscript after the authors performed the suggested corrections/additions.
Author Response
Thank you for your valuable comments that help improve the quality of our manuscript. Based on your comments, we have made appropriate revisions and responded point by point as follows:
Point 1: Although, I suggest including the specific approach for cancer treatments/therapy in the title.
Response 1: We have changed the title to “Extracellular Vesicles as Delivery Vehicles for Therapeutic Nucleic Acids in Cancer Gene Therapy: Progress and Challenges”.
Point 2: Addition of one table/image: Comparison as a summary (which are the advantages/disadvantages?) among the methodologies regarding to increase EV production, to improve nuclei acid drug loading efficiency, to extend circulation time, and to improve targeting capability.
Response 2: We have added advantages and disadvantages in Figure 2.
Point 3: Suggestion. Item: Clinical status of EV-based nucleic acid drug delivery systems: include one table summarizing all the information.
Response 3: We have added the Table 3 for this content.
Point 4: Which is the additional information the present review article has in comparison to Pharmaceutics. 2020; 12(10): 980. doi: 10.3390/pharmaceutics12100980. Extracellular Vesicle-Based Nucleic Acid Delivery: Current Advances and Future Perspectives in Cancer Therapeutic Strategies?
Response 4:
- Massaro, et al. (Pharmaceutics. 2020; 12(10): 980.) summarizes the methods for EV loading, parental cell-based engineering, EV surface functionalization, and their application in solid and hematologic tumors.
- We mainly discuss the advantages, challenges, and strategies to improve EV-based nucleic acids delivery from four aspects:how to increase EV production, how to improve loading efficiency, how to extend circulation, and how to improve tumor targeting. Apart from that, we systematically review EV-based delivery of nucleic acid drugs in the aspects of small nucleic acid drugs (including single-stranded antisense oligonucleotides, single-stranded miRNA, double-stranded siRNA), mRNAs, CRISPR/Cas 9 system, and the combinational delivery. These detailed classification and discussion are missing in C. Massaro, et al.
- For the clinical application perspective part, we list current status of EV-based nucleic acid drug delivery systems, including their trade name, source, cargos, and clinical trial status. We also discuss the challenges and perspectives of EV-based nucleic acid drug delivery systems for clinical application in these aspects: quality control, drug loading rate, stability, metabolism and dynamic tracking. These summarizing and discussion are not provided in C. Massaro, et al.
Point 5: Important to add the following reference: Extracell Vesicles Circ Nucleic Acids 2022; 3:14-30. doi: 10.20517/evcna.2021.21. Nucleic acid functionalized extracellular vesicles as promising therapeutic systems for nanomedicine.
Response 5: According to your kind comment, we have cited this paper, which is at [157] in revised manuscript.
Point 6: I would like to invite the authors to add the abbreviation list of words at the end of this manuscript.
Response 6: Thanks for your suggestion, we've added an abbreviated list of words at the end of the article, marked in red.
Round 2
Reviewer 1 Report
All concerns have been addressed.
Reviewer 2 Report
Manuscript revised as per suggested edits.